# Types of Resident and Price Distribution in Urban Areas: An Empirical Investigation in China Mainland

**DOI:** 10.3390/ijerph20010445

**Published:** 2022-12-27

**Authors:** Kaida Chen, Hanliang Lin, Fangxiao Cao, Xin Li, Shuying You, Qian Zhang

**Affiliations:** 1College of Landscape Architecture and Art, Fujian Agriculture and Forestry University, Fuzhou 350002, China; 2Department of Urban Planning, National Cheng Kung University, Tainan 70101, China; 3College of Civil Engineering and Architecture, Guangxi University, Nanning 530004, China; 4International Digital Economy College, Minjiang University, Fuzhou 350108, China; 5School of Economics and Management, Fuzhou University, Fuzhou 350025, China

**Keywords:** consumer behaviors, residence planning, spatial planning, developer types, housing prices, urban econometric

## Abstract

Finding an ideal residence in the city is a common and long-lasting topic for city residents. Therefore, understanding the value composition of urban residences and consumer preference can assist other future consumers in purchasing the appropriate residence in the appropriate urban location. Similarity, this information is helpful to municipal government planners in determining the use of urban land, to real estate developers in choosing where to develop commercial residences, and to the relevant research community in determining the effects of changes on the use of urban land. Although the study on housing prices influencing variables has long attracted scholarly attention, there has been limited research on the types of residences and developers, so it is essential to expand the research on this subject. In the study, Fuzhou, China, serves as the research context. The study employs econometrics to investigate the impact of residence and developer types on housing prices. Based on the study, it is shown that the price of commercial residences can vary depending on the types of residences and developers. The study also revealed that different types of residences and developers are subject to distinct levels of price regulation. In addition, it is found that different housing price impact variables have varying degrees of impact on different types of commercial residences and developers.

## 1. Introduction

Urban Housing prices have always been an unavoidable subject throughout the long history of human progress. People always have a significant interest in understanding the elements influencing housing prices due to the intricacy of their fluctuations and the range of angles from which they are viewed. In the United Kingdom, the United States, and Norway, where the real estate market has evolved more quickly, there has been more research on the prices of commercial residences, the main areas of study are the external influences on housing price and the characteristic factors of housing prices. Research on the external influencing factors of housing prices can be divided, according to scholars, into two aspects: demand factors and policy factors. In terms of the demand in the US real estate market, Poterba [1] contends that income is the most significant factor in determining housing prices. There is a negative association between a decline in interest rates and an increase in housing prices in a market economy [2]. Additionally, growth in employment in the UK might contribute to an increase in housing prices [3]. In Norway, population movements and structural changes can alter local income levels, which in turn affects housing prices [4]. In terms of policy issues, Mark and Goldberg [5], citing Vancouver, Canada as an example, suggest that the growth in land prices, which is driven by an unreasonable tax structure, influences consumers’ decisions to acquire a residence. The study of the characteristic factors of housing prices generally refers to the analysis of the composition of prices after the decomposition of residential commodity characteristics. Through an analysis of suburban residences in the United Kingdom, Garrod and Willis [6] determined that the average price of a residence increased by 7% for each additional room and by 6.9% for a garage. Clark and Herrin [7] evaluated the impact of school quality on housing prices in Fresn.

In cities where there is often a wide range of residence types, the transaction price of a residence often serves as a comprehensive overview of the consumer preferences in terms of residence types. Today, in addition to the basic function of living in a residence, there are many other additional ancillary functions, the value of which can be translated into the price of a residential transaction [8]. For different consumers, whose budgets and residential needs are not the same. How do consumers choose the right type of RESI and get the best value for money in terms of expected returns? In response to these inquiries, it is crucial to investigate the best advantages for consumers using an analysis of urban housing prices as a point of comparison.

Due to several policy interventions, there is a huge diversity of residence types across the globe, and the disparities in residence types have a significant effect on housing prices. Many researchers have analyzed the effect of different dwelling types on housing prices. A study conducted by Goetz, et al. [9] in Minneapolis, Minnesota, revealed that the presence of government-subsidized residences has a negligible impact on housing prices in the neighborhood. Galster, et al. [10] analyzed housing prices in several regions surrounding government-subsidized residence in Baltimore County, Maryland, and found that prices were lower the closer they were to the government-subsidized residence. Chen [11] argues that subsidized residences in China not only curb the rise in housing prices but also reduce the adverse effects of commercial housing price inflation. Gao [12] contends that affordable residences and public rental residences in China have a substantial effect on the expansion of consumption among middle- and lower-income groups. Albright, et al. [13] analyzed the effect of subsidized residences on housing prices in New Jersey, United States, and determined that the effect was minor. In Busan, South Korea, Kim, et al. [14] demonstrated that the proximity of public rental residences has a substantial effect on the pricing of adjoining commercial residences.

In a similar manner, the quality of the residence itself is evaluated differently based on the type of developer, which affects housing prices. Several scholars have investigated this topic. Due to its enormous demand and volume of transactions, China mainland’s real estate market is an excellent case study for examining the influence of residential type and developer type on housing prices [15]. In China, the concept of commercial residence originated in the 1980s and generally refers to the residence built and operated by certified real estate developers, including commercial residence, commercial-and-residential housing and business housing. During the period of the planned economy, Chinese citizens had poor earnings, so the government established a planned distribution system that incorporated social welfare and rent collection. A portion of the social welfare budget was utilized as a housing subsidy for citizens and the state committed funding for the construction of residences. The tenure-only residences were not owned by the residents, but they were expected to pay only token rent. China changed from a planned economy to a market economy between the 1980s and late 1990s, and a commercialized housing system was built to satisfy the needs of the overall economic reform. From the late 1990s to the present, China has clarified the dual qualities of urban housing as commodity and welfare, with the commodity attribute serving as the housing’s primary attribute and the latter as its supplementary attribute. By 2004, with the exception of a small number of public residences earmarked for a token rental by low-income households, the vast majority of public residence units eligible for sale had been purchased. With the conclusion of the public housing era, China established a government-subsidized residences system centered on affordable residence and public rental residence. There are numerous types of residences in China mainland, such as pure commercial residence, price-limited residence, affordable residence and private property residence. There are no definitive advantages or disadvantages among the many forms of residence, as they all possess their own unique traits. Among the numerous forms of residences in China mainland, owners of pure commercial residences have the land use right certificate, and the residences can be freely transferred, and are of higher quality. Generally, pure commercial residences are well-equipped with surrounding amenities, administered by dedicated property management, and have superior community security [16]. A price-limited residence is a type of residence with a limited price, type, and size. The affordable residence is a commercial residence with authority subsidies, which is economical and inexpensive but comes with the drawbacks of inconvenient transportation and transaction. A private property residence is a house constructed on communal land in rural areas without paying land premiums. Since the title deeds are not issued by the national housing authority, but rather by the township or village administration, the transaction is not legally protected. Self-built residences may not fulfill building construction safety regulations and pose a safety threat, but they are the least expensive to construct, making them the cheapest option for residents. For developer types, the categories can generally be classified by the attribute of the developer. In China’s early years, there were two primary types of real estate developers, one derived from housing management offices and the other from the reform of the logistic system of national organizations; both were essentially state-owned. In recent years, as a result of the continuous improvement of the market economy in China mainland and the formation of economic diversification, China’s real estate landscape has also undergone dramatic changes, with the primary investment of real estate shifting from state-owned enterprises to private and non-sole proprietorship enterprises. At present, China’s real estate developers can be categorized as sole proprietorship enterprise which contains central enterprises, state enterprises, private enterprises, foreign enterprises, and non-sole proprietorship enterprises. In addition to classification by attribute, developers can also be classified by their brand influence and origin.

In addition, different consumers have diverse housing requirements, which are frequently influenced by the types of residences and developers. From the perspective of different residence types, the price of a price-limited residence is lower than that of general commercial residence, and it is primarily for low- and middle-income families; affordable residence is moderately priced relative to the market price, making it accessible for low- and middle-income families; private property residence is located in towns and villages surrounding cities, with low prices, and is suitable for low-income groups and migrant workers. The school district policy of ‘enrollment in a nearby school’ has prompted many parents to seek out neighborhoods with high-quality schools for their children, resulting in the shortage of housing-reform residence within the high-quality school. For consumers considering the developer types, a poll reveals that more than half of consumers are interested in purchasing commercial residences built by developers with brand influence. The strength of brand developer helps prevent risks such as delayed delivery and interruption of construction becoming a terrible building due to a lack of funds. Brand developers have more ability and capital to invest in commercial facilities, landscape design, and other aspects of construction. Besides, brand developers can also introduce large-scale shopping plazas and other facility services [17]. In addition to this, consumers have a greater level of trust in local developers due to their underlying cultural familiarity and therefore prefer the brands of local developers when choosing the residence, when all other things are equal.

In China mainland, there are limited research perspectives relating to the type of residences and developers; therefore, it is important and necessary to investigate the impact of different types of non-commercial residences and various types of developers on housing prices in order to determine the optimal allocation of resources for society as a whole. The innovation of this study is its comparative analysis of the effects of commercial residence and developer types on housing prices, which is based on the urban setting of China mainland. The findings will assist anyone looking to buy a residence in selecting the residence and developer types that most closely matches their requirements and is most affordable. It is anticipated that the results will benefit all parties, including urban planning and administrative departments, housing developers, marketers and scholars, so that they can all gain valuable information from the research for their own industry and work together to promote the healthy development of living environment.

The sections of this paper are as follows: (i) a background introduction and study regression; (ii) a description of the methodology; (iii) the research data results, and (iv) the discussion and conclusions.

## 2. Literature Review

The previous research in China have focused on government-subsidized residences featuring the affordable residence. For instance, Liu and Chen [18] provide an overview of prior academic research on government-subsidized residences and a justification for how they influence housing prices. The study of developers’ effects on housing prices, on the other hand, is primarily concerned with state enterprises, foreign enterprises, and the influence of developer brands on housing prices.

### 2.1. Influence of Residence Type

The previous research in China have focused on the affordable residence, and mainstream scholars have argued that the affordable residence in China has had a restrain effect on the price of commercial residences. Wang and Zhao [19] examined the effect of subsidized residence construction on the Chinese housing market, proving that subsidized residences divert housing demand and offer lower prices, and the subsidized residence particularly affordable residence, is effective at curbing the price of the commercial residence. Wang and Gao [20] conclude that the development of government-subsidized residences in China has increased the total supply of housing and diverted part of the housing demand, hence stabilizing housing prices and rents. According to Zhao, et al. [21], the share of affordable residences dampens the price of commercial residences in Shenyang, China. An increase of 1% in the share of affordable residences in Shenyang, China, leads to a 0.08% reduction in the price of a commercial residence. In analyzing data from 29 provinces in China, Liu, et al. [22] concluded that both GDP per capita and property tax have a significant and positive impact on commercial residence prices, while the availability of affordable residences negatively affects commercial residence prices, and land prices have a significant but insignificant positive effect on commercial residence prices. In addition, they offer policy recommendations to reduce the price of commercial residences in terms of reasonable determination of affordable residence prices, enhancing the availability of affordable residences, changing the scope and substance of property tax levies, and reforming the land auction system. Li, et al. [23] reached similar conclusions by analyzing data on consumer demand for housing in the main city of Chongqing, China. Luo and Zhu [24] analyzed 964 residential neighborhoods in Shanghai, China, and discovered that the price of commercial residences within 0.5 km of affordable residences was 8.6% less than that of commercial residences 3.5 km away, and that the discount effect diminished with increasing distance. Additional research revealed that secondhand commercial residences in the neighborhood of affordable residences were less expensive than secondhand commercial residences in non-peripheral areas [25]. Zhang [26] believes that government-subsidized residences will lead to a decrease in the housing prices of commercial residences.

Nonetheless, other Chinese scholars have determined that the influence of affordable residences on urban housing prices is minimal. Xiao, et al. [27] found that affordable residence and commercial residence in China are fundamentally different because the consumers are separate; therefore, it is difficult to conclude that affordable residence is devastating to developers of commercial residence, and that the government has attempted to minimize the impact on commercial housing in its review and operation of affordable residence purchases. Chen and Wang [28] argue that subsidized residence should be based on addressing the housing needs of low- and middle-income groups and should remain relatively independent from the commercial residence market; consequently, the effect of subsidized residence on the price of commercial residence should not be anticipated. According to Fu [29], the availability of subsidized residences can reduce the price of commercial residence, but such effect in first-tier cities is minor.

However, many Chinese scholars believe that the widespread building of affordable residences will rise the price of commercial residences. The study by Wu and Li [30] in China shows that affordable residence increases national housing prices in the short run, but that this effect gradually increases, then decreases, and disappears after ten months. In addition, there is no long-term equilibrium relationship between affordable residence and national housing price growth, nor is there a Granger Causality between them. Zhou, et al. [31] study empirically the effect of crowding-out and diversion impacts of affordable residence supply on the prices of conventional commercial residence. The results indicate that affordable residence has a significant impact on the price of the general commercial residence in the long run, with the crowding-out effect being greater than the diversion effect, thus raising the price of general commercial residence; by region, the affordable residence has a greater influence on the price of a general commercial residence in economically developed regions, with the crowding-out effect being greater than the diversion effect. This justifies the following policy recommendation: the government should return the development of affordable residence to its original purpose of providing guaranteed residences, and should not use it as a tool to suppress housing prices when formulating real estate regulation and control policies. According to Zou, et al. [32], government-subsidized residences raise the market prices of commercial residences as a whole.

### 2.2. Influence of Developer Types

In prior research, a number of scholars have studied the impact of foreign developers on the price of commercial residences and found a positive correlation between the two. Liu [33] suggest that foreign developers have pushed up housing prices in China. Lan and Qian [34] empirically examine data on the commercial residence market in Xi’an, China, from 2000 to 2011 and reach the following conclusions: the greater the expectations of foreign developers for local commercial residence prices, the greater the increase in housing prices in the current phase; the greater the investment of foreign developers in the previous phase, the smaller the increase in housing prices in the current phase; and the greater the investment of foreign developers in production, the greater the increase in housing prices in the current phase. Gholipour [35] conducted a study on emerging economies and the role of foreign developers on housing prices appreciation was minimal in the countries studied. Gholipour, et al. [36] study in APEC countries further supports his view. Rong [37] also discovered through research that the reputation of foreign and Hong Kong developer brands increases housing prices.

In addition, research has revealed that Chinese state developers typically offer commercial residences at lower housing prices. For instance, Monkkonen, et al. [38] analyzed data on housing transactions in Chengdu, a medium-sized Chinese city, from 2004 to 2011, matching them to local amenities, public services, and developer type, and discovered that residences created by state enterprises were sold with a 7%-off discount. One reason for this, according to Monkkonen, Deng and Hu [38], is that state developers collaborate with local governments to provide affordable residences for citizens. Another reason could be that state developers are inefficient, and the low quality of property management and interior decoration may also contribute to lower housing prices.

A number of researchers have discovered that the developer’s brand is one of the most influential variables on housing prices. Roulac [39] empirically investigates the influence of brand, aesthetics, and practicality on housing prices by analyzing data from approximately 55 residences around the world, and shows that consumers generally prioritize developer brands when purchasing. In a study conducted in Hangzhou, China, Li and Chau [40] suggest that consumers are willing to pay a premium for developer brands, possibly because branded developers often sell finely-decorated residences. Rahadi, et al. [41] conducted a study on the factors influencing housing prices in the Jakarta metropolitan area in Indonesia and showed that there are four primary elements that influence developers’ pricing: design, reputation, amenities, and accessibility. Further research found that design, brand, reputation, amenities, reinvestment value, pricing policy, and speculative behavior were the factors influencing housing prices [42]. Rinchumpoo, et al. [43] found that the brand effect led to a 12.90% increase in housing prices in the Bangkok Metropolitan Region (BMR) sub-region of Thailand. Choi, et al. [44] analyzed the South Korean real estate market and discovered that developer branding has a premium effect on housing prices, and an additional study indicates that branded residential sales are greater than non-branded residential sales within the same location [45]. Chia, et al. [46] discovered that housing features, finance, distance, environment, and superstition affected consumers’ willingness to purchase a residence in Sabah, Malaysia. Using the example of middle-class housing in Semarang, Indonesia, Wijaya and Zulfa [47] demonstrate that the developer’s brand image has a major influence on the desire of the middle class to acquire a residence. Kim [48] found a premium on housing prices for location or area names in Seoul, South Korea.

## 3. Methods

### 3.1. Research Objectives

The study investigates the impact of residence and developer types on commercial housing prices in the China mainland, as well as the potential moderating variables for both variables. Fuzhou (119.28° E, 26.08° N.), the capital of Fujian Province, is selected as the reference city (Figure 1). It is an economically developed coastal region in the southeast of China and has a long history of urban development, having undergone all stages of housing development from the establishment of the People’s Republic of China. It also contains a wide range of residence and developer types and an adequate sample size, which makes it an invaluable reference for the study.

### 3.2. Framework

The study will investigate the two predetermined research objectives in the frame depicted in Figure 2. For the residences types objective, regressions will be conducted on a representative sample of all residences (1079). For the developer type objective, regressions will be run on a sample of pure commercial residences in residences, to rule out any policy-related or other-factor-related fluctuations in housing prices. The study commenced with a backwards-looking screening of the collected variables in order to find the housing prices influencing variables and the matching regression models. The final stable linear regression model serving as the standard model for the study was then determined by incorporating a number of typical residences type variables. The standard model is then used to investigate the influence of residence and developer type variables on housing prices. The interaction term variables are then added based on the standard model and the regressions that interact with the residence and developer type variables are studied and discussed based on the price effect variables identified in each study. The study concludes by conducting robustness tests on all regression models with significant study variables to confirm the consistency of the results.

In the study, regressions for the residence and developer type variables were undertaken from both an overall and a partial perspective in order to give a comprehensive examination of the correlation between the study variables and housing prices, after the standard model equations were defined. The regressions for all residences (1079) explain the extent to which the variable affects all residences (1079) as a whole, whereas the regressions for the partial samples explain the extent to which the variable affects the same type within its group. Regressions from both points of view are explored for both the linear regression models of the study variables and the interaction regression models.

### 3.3. Data

Through field research and online data gathering, the study collected panel data on second-hand residential transactions (1079) in Fuzhou City during the March of 2021. These data include information on the actual housing prices of the residential sample (explained variables) and the sample’s corresponding characteristics (explanatory variables). The three primary components of the characteristic variables are location environment, self-characteristics and facilities accessibility. The residence and developer type variables of this study are also included in the category of self-characteristics, which is detailed in Table 1.

To improve comprehension of the complex residence and developer types, this study establishes the links between the two types and depicts them in Figure 3. The number of the category name in the figure represents the samples of that type in this study. As stated in the introduction, there are three types of commercial housing in China mainland: commercial housing for commercial, commercial housing for residence and commercial housing for residence and commercial. However, the study focuses solely on commercial housing for residence (referred to in the paper as RESI, N = 1079). It can be subdivided into pure commercial residence (PCOM, N = 585) and non-pure commercial residence (NPCOM, N = 494), the NPCOM in this paper refer to residences including resettlement residence (RES), housing-reform residence (REF), public rental residence (REN), affordable residence (AFF), private property residence (PRO), price-limited residence (LIM) and tenure-only residence (TEN). Regarding developer types, the study divides the PCOM into developers established in the market economy period (MCOM, N = 566) and established in the planned economy period (PLCOM). The MCOM can also be divided into two parts: sole proprietorship developers (SPro) and the NSCOM developers (NSCOM, N = 352). Developers of the SPro are divided into four types, namely central enterprise developers, state enterprise developers, private enterprise developers and foreign enterprise developers. The NSCOM are divided into two attributes: listed and non-listed developers, this is the most sophisticated classification covered in the study. For developer types, the categories can generally be classified by the attribute, the brand influence (T500) based on the China Top 500 Real Estate Rankings 2021 and the origin (MIN) based on whether the developer originated in Fujian Province. For this study, the distribution of the residence and developer types in the Fuzhou region is shown in Figure 4.

### 3.4. Methodology

This study applies some of the modelling methods of econometrics. This comprises the Hedonic price model as the fundamental model, the stepwise regression strategy for variable screening, and the econometric interaction model for identifying moderating effects.

#### 3.4.1. Hedonic Price Model

The Hedonic Price Model is a linear model function utilized extensively in real estate and other fields involving land value, price appraisal, and price forecasting [49]. It comes in a variety of forms, such as linear, semi-logarithmic, and double-logarithmic. However, the double logarithmic model more accurately depicts the existence of a considerable marginal utility of the transaction price for the attributes of a property when acquiring a residence, rendering the simulation process more realistic and reasonable. Consequently, the pricing model utilized in this research is a double-logarithmic model, which differs from the model utilized by scholars in the past, and the link between the housing price and the explanatory variables are depicted in Equation (1).
(1)$$\ln\mathrm{Price}\;=+\overset{i\;=\;n}{\underset{i\;=1}{\sum }}\beta_{k}\ln x_{\mathrm{ki}}+\varepsilon_{i}{}_{}$$
where Price in Price(i) represents the housing price, i represents the data in the i-th property, and Price(i) represents the average price per square meter in the i property, which includes the sum of the housing price itself, the cost of decoration, and other aspects of the price, i.e., the average transaction price of the i property; where k is the characteristic factor, it represents the k-th characteristic influence factor among many attributes, i also represents the i-th property; n represents the data performance of the n-th characteristic influence in the i property; $\gamma_{k}$ represents the unstandardized coefficient of the k-th relevant characteristic influence on housing price; and $\varepsilon_{i}$ represents the stochastic error term. In addition to the coefficients contained within the model, regression analysis permits the estimation of the significance of the principal positive or negative explanatory variables influencing housing prices.

#### 3.4.2. Stepwise Regression Model

In linear regression models, stepwise regression is used to select independent variables [50]. Essentially, variables are introduced sequentially when their partial regression sum of squares is empirically significant. Upon introducing each new variable, each existing variable included in the regression model is checked one by one, and those considered insignificant are eliminated in order to ensure that each variable in the resultant subset of independent variables are significant.
(2)$$F_{j}^{\left(k-1\right)}=\;\min\left\{F_{1}^{\left(k-1\right)},F_{2}^{\left(k-1\right)},\cdots ,F_{k-1}^{\left(k-1\right)}\right\}$$

The procedure is repeated until no additional variables can be introduced. At this point, all factors in the regression model are significant for the dependent variable. The stepwise regression approach consists of both a forward and a backward method. The backward method is selecting a smaller number of independent variables and eliminating them one by one until there are none left to eliminate, with the specific formula model as shown in Equation (2).

#### 3.4.3. Econometric Interaction Model

In econometrics, the inclusion of an interaction term in a linear regression model is a special treatment of a kind of regression equation model in which the interaction can be viewed as the outcome of the interaction between two or more influential elements. This method helps expand the explanatory viewpoint and depth of explanatory factors that are mutually influenced by many explanatory variables in the regression model. During the course of the research, both additive and multiplicative interaction terms were considered, but after comparing the significance of the relevant data, the multiplicative interaction regression model with better fit and significance was selected as the method for this interaction study, with the specific formula model as shown in Equation (3).
(3)$$y_{i}\;=+\beta_{m}{}_{}+x_{\mathrm{ni}}+\beta_{\mathrm{mn}}x_{\mathrm{mi}}x_{\mathrm{ni}}+\varepsilon_{i}$$
(4)$$\frac{\beta_{1}{}^{2}y_{i}}{\beta_{1}x_{\mathrm{mi}}{\beta x}_{\mathrm{ni}}}=\frac{\beta_{1}\left(\frac{\beta_{1}y_{i}}{\beta_{1}x_{\mathrm{mi}}}\right)}{\beta_{1}x_{\mathrm{ni}}}={\;\beta }_{\mathrm{mn}}$$

In the Interactive Regression Model, Equation (4) represents the ‘interaction effect’ which is the correlation between the effect of an explanatory variable and its magnitude.

## 4. Analysis Result

There will be two sections for presenting the study’s data results: one for linear regression and the other for interaction regression. Along with the data results, the related data interpretation will also be presented.

### 4.1. Linear Regression

#### 4.1.1. General Variables

In the study, the regression results were achieved by gathering the variables that may influence the housing prices, as indicated in Table 2 and the regression equation was optimised by screening the effective influencing variables using the stepwise regression approach. The regression process of variable screening was separated into eight steps using the backward method to eliminate non-significant factors and identify the variables with a substantial impact on the sample housing prices, as well as the best-fitting model (BFM).

As the greening rate in the residential characteristic variables are strongly correlated with residence and developer types, the study continues to examine the subsequent regression fitting process by excluding the greening rate variable from the model function to reduce the co-linear interference in the regressions for residence and developer types. The greening rate of residential communities in China is often closely correlated with the residences and developer types; hence, the study may be viewed as part of the influence of the residences and developer types on housing prices when analyzing relevant topics.

The study will present the model’s regression results for the difference in the number of regression samples from two metric viewpoints for the influence of residence and developer types on housing prices. The first metric perspective represents the regression results for the RESI, whereas the second metric perspective represents the regression results for the RESI after screening and classification. In addition, to ensure the credibility of the sample variables, the quantity of valid data after screening and classification for the regression sample must be more than 300.

#### 4.1.2. Residence Types

The effects of residence types on housing prices are seen in Table 3. In the regressions of the influence of residence types on housing prices, the study first examines the RESI, from the types of the PCOM, the RES, the REF, the REN, the AFF, the PRO, the LIM, and the TEN. Secondly, it investigates the NPCOM, from the types of the RES, the REF, the REN, the AFF, the PRO, the LIM and the TEN.

The following is a summary of the specific regression results regarding the effect of residence types on housing prices. All regression equations with an R-squared larger than 0.659 indicate a strong fit and the findings are explanatory. In each regression model for the RESI, the presence of the PCOM variable has a substantial positive influence on housing prices, whereas the presence of the RES variable has a significant negative influence on housing prices. Besides, the price of the TEN was considerably higher than the price of other NPCOM in the individual regression models for the sample of the NPCOM.

#### 4.1.3. Developer Types

The results for the impact of developer types on housing prices are presented below (Table 4). In the regressions on the impact of developer types on housing prices, the study first discusses the developer’s Top 500 brand influence, origin and attributes (MEco, SPro, Lis) on housing prices for the PCOM. Partial regressions are then conducted for the sub-samples, with the regressions discussing the developer’s Top 500 brand influence, origin and attributes (SPro) on housing prices for the MCOM. The impact of Top 500 brand influence, origin and attributes (listed) on housing prices are discussed separately for the NSCOM.

The specific regression results for the effect of developer types on housing prices can be summarised as follows. The overall R-squared of all the regression equations is greater than 0.620, which is a good fit and the findings are explanatory. The results of the regressions for the PCOM show that there is a significant positive correlation between Top 500 brand influence and origin on housing prices. The Top 500 brand influence and the origin variables are also positively correlated for commercial residences for the MCOM, while whether the residences are the SPro has no effect on the housing prices. For the NSCOM, only the Top 500 brand influence is positively significant.

### 4.2. Interactive Regression

#### 4.2.1. Residence Types

The findings of the interaction regressions between the residence types and general variables on housing prices can be found in Table 5 and Table 6. In the interaction regressions between the residences types and the general variables on housing prices, the study first examines the RESI, there are the PCOM, the RES, the REF, the REN, the AFF, the PRO, the LIM, the TEN and general variables have the moderating influence on housing prices. This is followed by a consideration of the RES, the REF, the REN, the AFF, the PRO, the LIM, the TEN and general variables that have a moderating influence on housing prices for the NPCOM.

The following is a summary of the regression results for the interaction of residence types with other general variables influencing housing prices. The R–squared of the overall interaction regression equations whose fits were all greater than 0.659 was a suitable fit and the findings are explanatory. As for the RESI, the interaction of PCOM with variables of self–characteristics-residences with a high-quality primary school, residences age and management fee. At the same time, the location environment variables-regional GDP level. The facilities’ accessibility variables–the distance to the closest green space, dump and funeral facility were significant when it came to the influence on housing prices, while with a high-quality middle school and the distance to the closest main water source were only marginally so. A significant effect is observed in housing prices when the RES interact with variables of self-characteristics-with a high-quality primary school and middle school as well as residences management fee, while the facilities accessibility variables–the distance to the closest dump, rail stations, hospitals and funeral facility have marginally significant effects on housing prices. The REF and regional GDP levels have significant interaction. Housing prices are significantly influenced by the interaction of the REN and the location environment variables-located inside the second ring road, regional GDP level, the density of population, self-characteristics variables-with a high-quality primary school, facilities accessibility variables–the distance to the closest rail station. There is a marginally significant interaction between the TEN and the self-characteristics variables-residences management fee. As for the interaction regressions for the NPCOM, the interaction between the RES and variables of facilities accessibility–the distance to the closest rail station, hospital, and green space is significant for housing prices, while interactions with the residences management fee, quantity of population, located inside second ring road and third ring road are marginally significant. The REF is significantly influenced by regional GDP level, while the quantity of population, located inside the third ring road and the distance to the closest green space is marginally influenced. The interaction between the REN and variables of the location environment-located inside the second ring road, regional GDP levels, the quantity of population, the density of population, variables of self-characteristics-with a high-quality primary school and variables of the facilities accessibility–the distance to the closest rail station has a substantial influence on housing prices. The association between the distance to the closest dump and the PRO is marginally significant.

#### 4.2.2. Developer Types

The regression results for the interaction between developer types and other general variables on housing prices are shown in Table 7. In the interaction regressions between developer types and general variables on housing prices, the study first discusses whether there is an interaction between the developer’s Top 500 brand influence (T500), origin (MIN) and attributes (MEco, SPro, Lis) and general variables that moderate the influence on housing prices for a sample of the PCOM. Afterwards, the developer’s Top 500 brand influence, origin, types and general variables on housing prices are discussed separately for the MCOM and the NSCOM.

The findings of the interaction regression between developer types and inherent variables are presented in Table 8. In the interaction regressions between developer types and their price influence variables, the study constructed two-by-two interaction terms for three characteristics: the developer’s Top 500 brand influence (T500), origin (MIN) and attributes (MEco, SPro, Lis). Similar to the previous sample groupings for the interaction between developer type and other general variables on housing prices, the discussion begins with the PCOM, and then moves on to the MCOM and the NSCOM.Taking into account the interaction between developer types and other general variables on housing prices, the following regression results can be summarized. The R-squared of all the interaction regression equations whose fit is greater than 0.621 is a good fit and the conclusions are explicative. For the interaction regressions of the PCOM; there was a significant influence on housing prices for the interaction between the developer’s originating location and the self-characteristics variables-with a high-quality primary school and the facilities accessibility variables–the distance to the closest rail station and main water sources. Housing prices are significantly influenced by the interaction between the influence of the developer and the self-characteristics variable-residences management fee, the interaction with whether the residences were built after 2000 is marginally significant. Housing prices are significantly influenced by the interaction between whether the developer is the SPro and the self-characteristics variable-with a high-quality primary school. Depending on the distance to the closest main water source and dump, the developer is established in the market economy period may have a marginally significant influence on housing prices. For the interaction regressions of the MCOM, there is a significant influence on housing prices for the interaction of the developer’s origin with the self-characteristics variables-with a high-quality primary school and the facilities accessibility variable–the distance to the closest main water source; as well as a marginally significant influence on housing prices for the interaction with the distance to the rail station. Housing prices are significantly influenced by the interaction between the influence of the developer and the self-characteristics variable-residences management fee; the interaction with whether the residences were built after 2000 has only a marginal effect. The interaction between whether the developer is the SPro and the self-characteristics variable-with a high-quality primary school has a significant influence on housing prices. For the NSCOM; there is a significant influence on housing prices for the interaction with the self-characteristics variable-residences management fee; and a marginally significant influence on housing prices for the interaction with the distance to the funeral facility.

Based on the regression results, the interactions between developer types variables are summarized as follows. Those interaction regression equations with an R-squared greater than 0.619 have a good fit and are potentially explanatory. For the sample of the PCOM: the interaction between whether the developer is established in the market economy period, the China’s Top 500 real estates and the origin from Fujian Province is significant for housing prices; the interaction between whether the developer is the SPro and the Top 500 brand influence of the developer is significant for housing prices; the interaction between the developer’s Top 500 brand influence and origin is marginally significant for housing prices. For the MCOM, the interaction between whether the developer is the SPro and a Top 500 brand is significant for the housing prices, whereas the interaction between the developer’s Top 500 brand influence and origin is marginally significant for the housing prices.

### 4.3. Robustness Test

The study conducted a sample robustness regression for regression models that have significant interactions in order to verify the stability of the interaction regression results. The results of the above study were tested for stability with Table 9, Table 10 and Table 11, which contain some of the regression models that passed the robustness test. Robustness regressions consist of a linear regression section and an interaction regression section. The tables in the linear regression section cover regression tests for residence and developer types, whereas the tables in the interaction regression section cover regression tests for the interaction of residence types with general variables, developer types with general variables and developer types themselves.

## 5. Discussion

### 5.1. Linear Regression

This element of the study discussion will be broken up into two sections, one for the linear regression segment and the other for the interaction regression section. The topic of linear regression will cover the outcomes of the linear impacts of the research variables and general variables. The results of the moderating effects of the control factors on the study variables will be given in the interactive regression section.

#### 5.1.1. Residence Types

An explanation of the reasons for the impact of residence type variables on housing prices can be developed as follows (Table 3). For the regressions of the RESI, the PCOM presents a significant positive influence on housing prices, while the RES present a negative influence on housing prices. In other words, PCOM will have a price proportional to the value component associated with them, as purely marketable commodities. For reasons related to their policy orientation, the lack of quality of the residences and the low quality of the neighbours, the price of the RES in the Chinese market is significantly lower than the price of commercial residences. Based on the regressions conducted on the NPCOM, the study concludes that the TEN have lower prices than the rest of the NPCOM. It is well worth analysing and discussing the fact that their prices will be higher than those of other NPCOM. The study concludes that, although the neighborhood contains the TEN, the RESI sold are normal ownership residences in the neighborhood. Additionally, there are other positive value-added characteristics that are advantageous in this sample, such as developer influence, or other facilities in the neighbourhood, thus creating a statistical misjudgment of the results and yielding a result that the TEN are more valuable than other NPCOM.

#### 5.1.2. General Variables

Through the variable screening process (Table 2), the study identified significant variables, insignificant variables (colinearity) and the direction of significant variables on the impact of housing prices. The significant variables consist of the self-characteristics variables-whether the residences were built after 2000, with high-quality primary or middle school or not, residences management fee and greening rate; the location environment variables-whether located inside second and third ring road or not, the quantity of population, the density of population and the level of regional GDP level; and the facilities accessibility variables–the distance to the closest rail station, closest hospital, green space, main water source as well as dump and funeral facility. The insignificant and colinearity variables contain variables of the self-characteristics variables-the quantity of employed population (strong colinearity with the quantity of population and regional GDP levels); location environment variables-whether the residences are high-rise buildings or not, the density of building (not significant with the effect of housing prices); and the facilities accessibility variables–the distance to the closest market, scenic spot, factory, gas station (not significant with the effect of housing prices). The positively correlated variables are the residences with high-quality primary and middle schools, management fee, greening rate, whether located inside the second and the third ring road or not, the quantity of population, the density of population and regional GDP level. The facilities’ accessibility variables-the closest rail station, hospital, green space, and main water source; the negatively correlated variables are whether the residences were built after 2000 and the accessibility of dump and funeral facility. The variables that influence housing prices obtained above are largely consistent with the results of prior research and reflect the physical and psychological needs of customers for the residences. The reasons why customers prefer residences in the center part of the city with more employment opportunities (reflected by regional GDP level, quantity of population and density of population), high-quality residential communities (with the high quality primary or middle school, high service quality of property management), more facilities accessibility and away from dump and funeral facilities are readily apparent and will not be elaborated upon.

#### 5.1.3. Developer Types

The other variable of interest discussed in the study, the developer types variable, can be discussed in terms of its influence on housing prices as follows (Table 4). According to the regression analysis of the PCOM, developers’ Top 500 brand influence and origin have a positive influence on housing prices. Developers with a strong brand presence, namely in the T500 and the MIN are able to increase housing prices. The consumer recognizes the influence of developers and prefers to deal with the MIN. In contrast, the results of the regression for the sample of the NSCOM only showed consumer recognition of the Top 500 brand. Therefore, to summarise the above research, it can be found that regardless of the type of commercial residence, the developer brand has a significant influence on consumers in purchasing residences, with the greater the Top 500 brand influence, the higher the housing prices.

### 5.2. Interactive Regression

#### 5.2.1. Residence Types

Analysis of the interaction regression results (Table 5 and Table 6) for the residence types variables and other general variables yielded the following discussion. The exploration consists of regression for the RESI and the NPCOM in Figure 5. A moderating effect was found in the impact of the explanatory variables on the price of the RESI, regardless of whether they were the PCOM, the RES, the REF, or the REN. As for the regression results of the NPCOM, transactions, in addition to similar results found in the overall regression, additional moderating effects on housing prices were found for the RES and the REN.

Firstly, we discuss whether the residence is PCOM. It is important to note that when the residence is the PCOM, the better the economic base of the area, the less the difference between the housing prices and whether it is the PCOM, although the PCOM is always more expensive than the NPCOM. The NPCOM premiums are higher with a high-quality primary school than for the PCOM, i.e., it is likely that the NPCOM will be aligned with a high-quality primary school than the PCOM. The PCOM residences management fees tend to increase more rapidly in line with housing prices, whereas the NPCOM residences management fees have a more significant marginal influence on housing prices, i.e., Commercial residences are more sensitive to increases or decreases in residences management fees than the NPCOM. The PCOM appears to be more resilient to the impact of the funeral facility on housing prices than the NPCOM, possibly due to the fact that the PCOM is sited to some extent as far away from the funeral facility as possible. In a similar manner, the NPCOM does not appear to be as resistant to the impact of dump on housing prices as the PCOM. For the PCOM, the green space accessibility is more likely to increase prices than for the NPCOM. The NPCOM housing prices are not influenced by the green space accessibility, indicating that consumers buying the PCOM are willing to pay more for green space accessibility. Being the PCOM has a negligible impact on the price of the residences built before 2000, but for the residences built after 2000, being the PCOM can raise the housing prices more.

If the residence is not the PCOM, the following will occur. When a residence is the NPCOM with a high-quality primary school, the housing prices of the RES with a high-quality primary school is higher than the NPCOM with a high-quality primary school, which means a high-quality primary school has a greater appreciation for the RES. The results are similarly consistent for the RES with a high-quality middle school. At the same time, its residence management fees do not reflect well the trend in housing prices increase compared to commercial residences that do not include the RES. The RES has a stronger marginal effect on residence management fees than commercial residences without the RES. RES are also more vulnerable to the negative impact of the closest dump on housing prices. The higher the regional GDP level where the REN and the REF are located, the faster the increase in housing prices. This indicates that the REN and the REF possess additional subsidiary values which are influenced by the regional GDP level. Furthermore, the quality of the living environment of the REN is inversely proportional to the density of the population of the area; the higher the density of the population, the lower the housing prices. Additionally, the housing prices decrease when it is closer to the railway station, which may reflect the quality of the living environment, which includes the REN, which decreases when it gets closer to the railway station. Further, the quality of the living environment in the REN is inversely proportional to their centrality. This results in the lower price for housing located inside the second ring road and the higher price for residences outside the second ring road. Lastly, the REN is less expensive when it is with a high-quality primary school. 

In addition to a discussion of the results consistent with the regressions for the RESI, additional discussions are provided below for the regressions for the NPCOM. As a first point, the housing prices in the NPCOM that include the RES are more influenced by the accessibility of the rail station than those in other the NPCOM. In general, the closer the rail station is, the higher the housing prices; however, for the NPCOM which does not include the RES, the price does not significantly differ regardless of whether the residences are close to the rail station. Additionally, the NPCOM including the RES are more susceptible to hospital accessibility than other the NPCOM, i.e., the closer the hospital is to the residences, the higher the price increase; however, commercial residences without the RES will have virtually no fluctuation in price. Furthermore, the NPCOM including the REN will exhibit a large quantity of population size dependency than other NPCOM in terms of housing prices, i.e., the larger the population size, the higher the housing prices increase; however, the NPCOM excluding the REN maintain relatively stable prices in response to population changes.

#### 5.2.2. Developer Types

Based on the results of the interaction regression between the real estate variables and other general variables (Table 7 and Table 8), the interaction model corresponding to the findings can be obtained (Figure 6) and the following analysis and discussion can be developed. Below is a discussion of common data results from the three survey subgroups: the PCOM, the MCOM and the NSCOM. For the PCOM developers, the price of a non-Local developer’s residence decreases when it is with a high-quality primary school, while the MCOM prices significantly increase when it is with a high-quality primary school, reflecting the establishment in the market economy has a very high price premium for high-quality primary school commercial residences with Local origins. Meanwhile, MIN rely more on rail accessibility to boost their own prices and are consistently higher than the housing prices of the non-MIN. The residences built by the non-MIN are closer to the main water source and will be accompanied by higher prices for the closest main water source. If the PCOM developer is a China’s Top 500 real estate, the housing prices rise more quickly with the quality of the residences management fee and this effect is even greater for China’s Top 500 real estate’s MCOM. As a result, China’s Top 500 real estates are able to generate more revenue from the residence management fees. For the PCOM developers, the increase with a high-quality primary school benefits is stronger for residences of the SPro than for residences of the NSCOM developers.

The following discussion can be developed based on the coefficient characteristics of the interaction regression between the real estate variables. The discussion again focuses on three subgroups, the PCOM, the MCOM, the NSCOM and the only case with a housing price moderating effect is found. In particular, for the PCOM, when the developer of the MCOM is China’s Top 500 real estates, the Top 500 brand influence contributes to a significant increase in housing prices. This may reflect consumer recognition of the value of the developer’s Top 500 brand influence.

## 6. Implication and Conclusions

### 6.1. Implication

To summarise the data findings of the above study, the relationship between housing prices and spatial distribution is spatially visualized as Figure 7. Figure 7 present the geographic location of residence and developer types, in relation to other facilities. The redder the area in the graph the higher the housing prices for the available types of residences in the area, the bluer the lower the housing prices.

Combining the graphic results, for consumers, the following three recommendations are provided. The first is for consumers who have no specific objectives. Those who are concerned about a high-quality primary school may choose to purchase the NPCOM with a high-quality primary school; if consumers wish to purchase residences of value without high residences management fees can purchase the NPCOM; if consumers have a limited budget and not cared about close to dump and the funeral facility can choose the NPCOM close to dumping and funeral facility; if green space is a priority, consumers can purchase the NPCOM that do not include the RES close to green space; if consumers desire the residences built after 2000, he or she can choose the NPCOM; if the consumers values the PCOM quality, the residences built before 2000 would be a wise choice. The above options provide maximum demand benefits at a minimal cost. In addition, if consumers with limited budgets are only looking for high-quality primary or middle school, they can opt to purchase residences with a primary or middle school that do not include the RES in order to be more cost-effective. Those who want residences with the lower residences management fee, younger consumers who are less in need of hospital accessibility or rail accessibility can acquire the RES to fulfil maximal expectations (Figure 7b). Alternatively, low-income groups with the high demand for rail accessibility, living inside the second ring road or high-quality primary school may also purchase the REN with the lower regional GDP level or higher density of population in order to reduce expenses (Figure 7c). Finally, consumers who plan to purchase the PCOM, who require high-quality primary school can purchase the residences from the local developer established in the planned economy period or the NSCOM developer. For those who require rail accessibility and prefer a waterfront environment or a river view (Figure 7d), they can purchase the residences from a non-local developer and get a more cost-effective return. If the consumers are looking for real estate value with the lower residence management fees, consider purchasing the residences without China’s Top 500 brand influence.

For the government, the following suggestions can be made for the planning of residence types (Figure 7a). The NPCOM can choose sites away from dump and funeral facilities to increase housing prices, especially the RES can choose sites close to the rail station, hospital, green space and away from dump and funeral facilities to increase its housing prices (Figure 7b). In addition, to increase the value of the REN, the planning department can implement policies to moderately relocate the REN who are currently living in urban areas to the city’s suburbs and to plan the REN outside the second ring road with the high regional GDP level, high quantity of population and low density of population. Additionally, the REN can choose sites close to rail stations in land function allocation planning to boost its utilisation value (Figure 7c). In addition, for the Chinese government, it should guide the direction of reform and opening up, integrate into the market economy survival environment, improve market competitiveness and promote social economic development.

For developers, the following recommendations are based on the findings of the Top 500 brand influence and origin studies. Developers of the PCOM can choose land near dump and funeral facilities to get lower land costs and housing prices closer to the neighbouring areas, developers of the RES can choose sites close to hospitals and rail stations; developers of the REN can select sites with a high quantity of population. All of the above can yield higher real estate profit margins. Developers origined from Fujian may choose to bid for land parcels closer to the rail station and the main source of water, as consumers seem to be more willing to pay higher housing prices for residences with rail accessibility and river view from local developers (Figure 7d). Furthermore, developers should be more competitive and become China’s Top 500 real estates so that they can gain consumer recognition and therefore generate larger revenue from housing prices and residence management fees.

For researchers, the process review and summary of the study can be concluded. According to the above study, it is well worth exploring the reasons for and composition of the other values attached to the regional economic base of the REN and the REF. A more comprehensive model of the impact of housing prices may be possible by taking into account other factors that affect housing prices. At the same time, scholars can continue to test the results in space using spatial regression models, such as SEM or GWR models, in conjunction with appropriate spatial weighting matrices, based on the results of the current phase of research.

### 6.2. Conclusions

There are many factors that affect housing prices, which can be classified into three main variables: location environment, self-characteristics and facilities accessibility. Housing prices are influenced by all of these variables, depending on consumer needs and psychological preferences. Previous studies have confirmed the general influencing variables in the study.

Different types of residences and developers have a great influence on housing prices. Housing prices are positively influenced by the PCOM, whereas housing prices are negatively influenced by the RES. The NPCOM and the TEN have a significant negative influence on housing prices, which is highly relevant to China’s actual conditions and policy. As far as developers are concerned, there is a positive influence on housing prices due to the developer’s Top 500 brand influence (T500) and origin (MIN).

Various housing price impact variables have differing degrees of price impact moderation for different types of residences and developers. Whether the residences is the the PCOM, the RES, the REF or the REN can have reciprocal moderating effects on housing prices. Similarly, developers’ Top 500 brand influence, origin and attributes also moderate the effect of other variables on housing prices. This allows different groups of consumers to choose different types of residences and developers according to their needs in order to maximise the benefits of their objectives.

### 6.3. Limitation

Even though the current study examined the correlation between residence and developer types on housing prices, it had a number of limitations. First and foremost, the collection of factors influencing price variables in the area of housing prices research topics is not sufficient to allow for a better explanation of the fitted regression functions. Additionally, residence and developer types tend to correlate with other price-influencing variables, for instance, residence types correlate with the facilities accessibility, whereas developer types correlate with the greenery of residential communities. Consequently, the impact of the residence and developer type variables on price alone would also be fully objective. Additionally, because residential communities are defined as a whole, the sample size of a city is often too small and a more refined study of residence and developer types is not available, thus leaving the depth of the study to be improved. Last but not least, the lack of accurate developer information in some residential communities has resulted in a hierarchy of developer types based on a minimum classification of whether they are listed or not. The overall stratification has not been refined or refined completely. It is hoped that future scholars and researchers will be aware of these issues in advance and will continue to conduct research on the impact of residence and developer types on housing prices.

## Figures and Tables

**Figure 1 ijerph-20-00445-f001:**
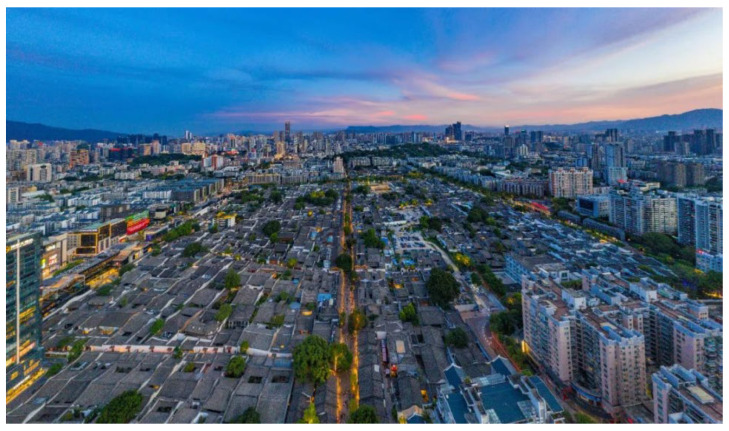
Aerial view of the city proper of Fuzhou.

**Figure 2 ijerph-20-00445-f002:**
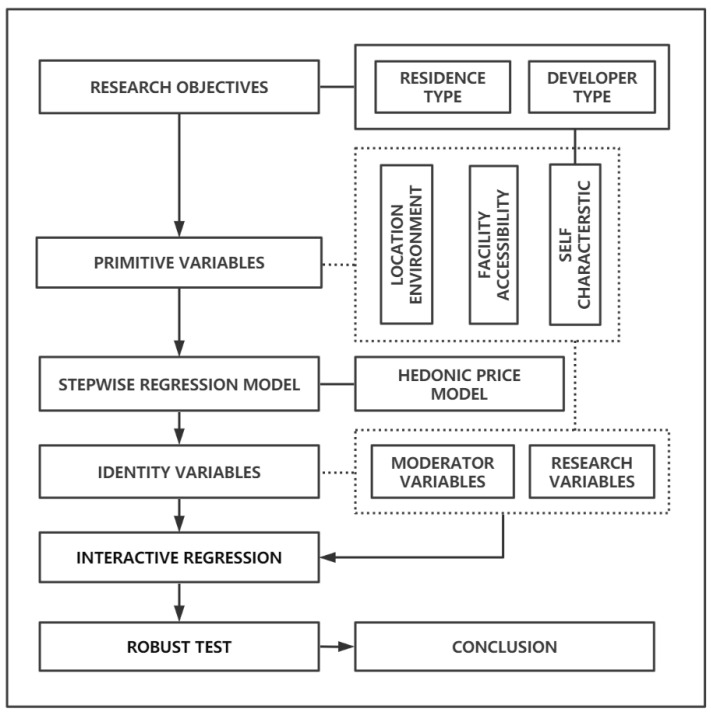
Frame diagram.

**Figure 3 ijerph-20-00445-f003:**
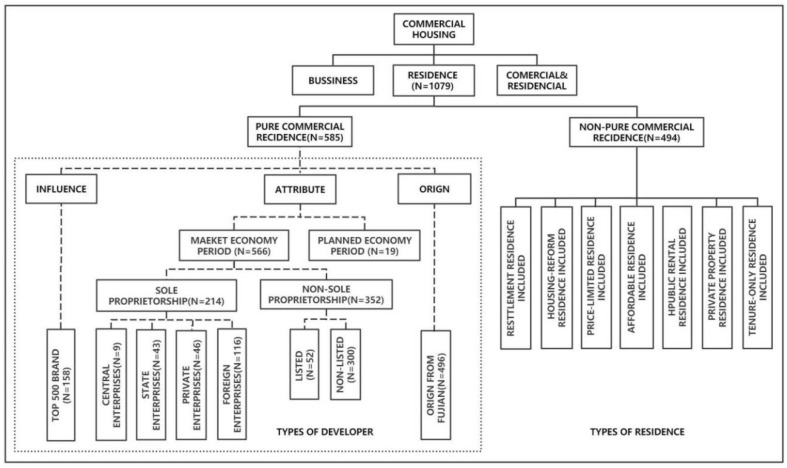
Relationship of residence & developer types variables in China.

**Figure 4 ijerph-20-00445-f004:**
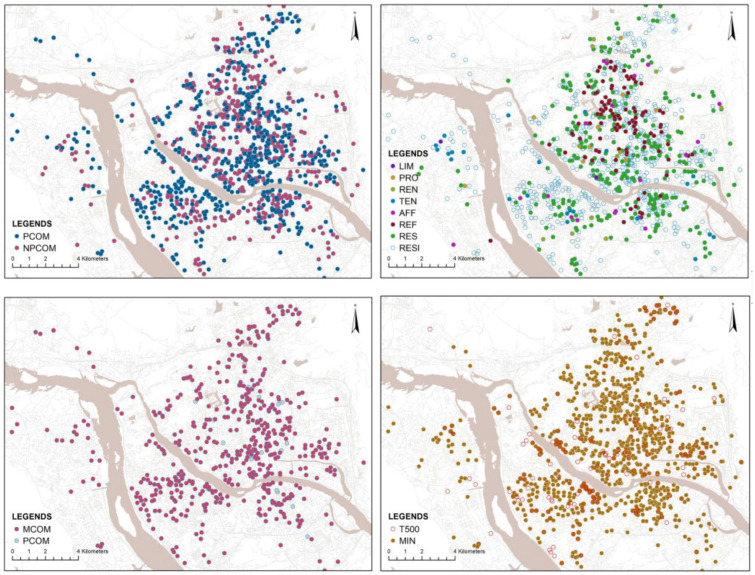
The current distribution of different types of residences and developers.

**Figure 5 ijerph-20-00445-f005:**
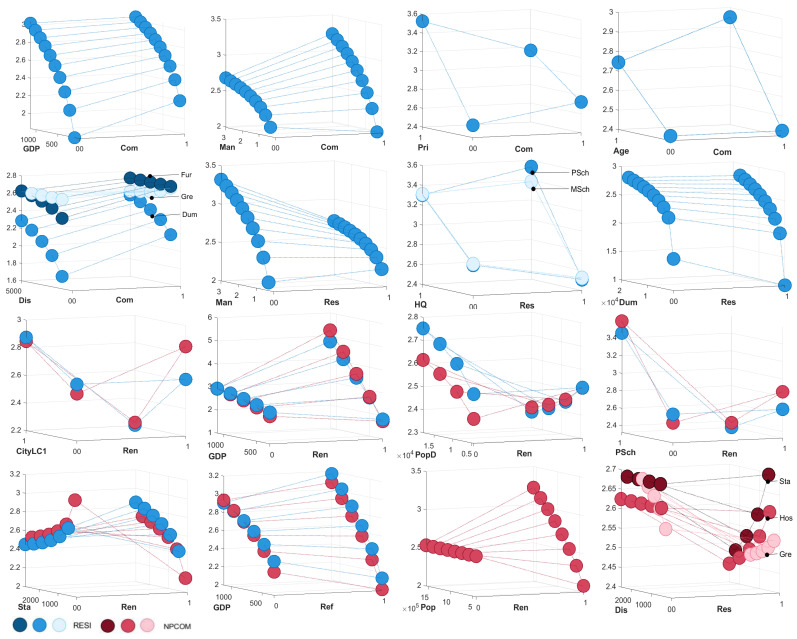
Price model chart of residence types.

**Figure 6 ijerph-20-00445-f006:**
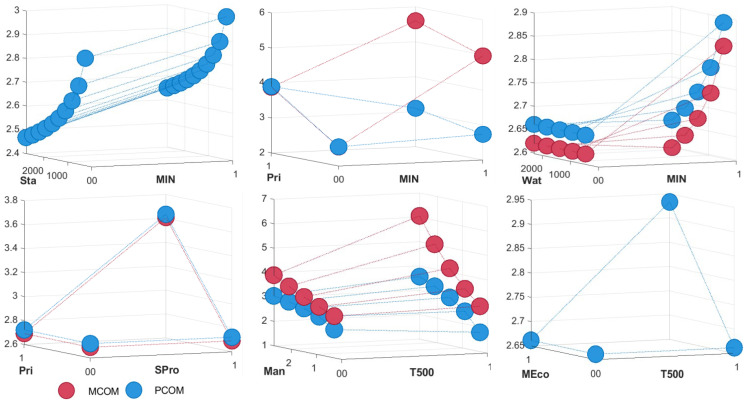
Price model chart of developer types.

**Figure 7 ijerph-20-00445-f007:**
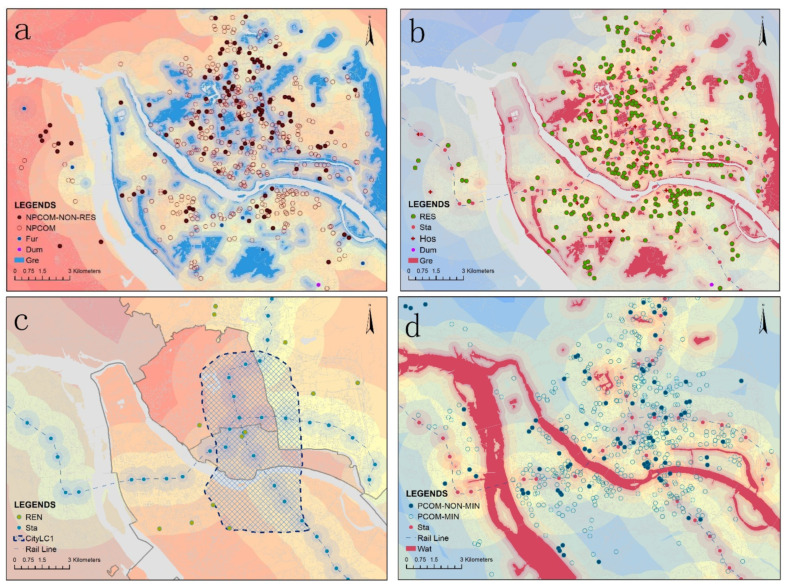
Relationship between housing prices and land space. (**a**) NPCOM, (**b**) RES, (**c**) REN, (**d**) PCOM.

**Table 1 ijerph-20-00445-t001:** Descriptive statistics of the variables (N = 1079).

Description		Mean	Expect Sign
Outcome Variable			
Pri	Housing prices (ten-thousand yuan/m^2^)	2.810	
Explanatory Variables			
Int	Interactive Variable	-	
Variables of the Location Environment			
CityLC1	Dummy variable, 1 for the residence inside Second Ring Road, 0 else	-	+
CityLC2	Dummy variable, 1 for the residence inside Third Ring Road, 0 else	-	+
Pop	Quantity of population in March 2021 (n)	819,222.390	+
Pop	Density of population in March 2021 (person/km^2^)	11,173.178	+
EPop	Quantity of employed population in March 2021 (n)	24.930	+
GDP	Per capita GDP in March 2021 (hundred million yuan)	567.270	+
Variables of the Self-characteristics			
Age	Dummy variable, 1 for the residence built after 2000, 0 else	-	+
PSch	Dummy variable, 1 for the residence with high-quality primary school, 0 else	-	+
MSch	Dummy variable, 1 for the residence with high-quality middle school, 0 else	-	+
Flo	Dummy variable, 1 for the residence is high-rise building, 0 else	-	-
BuiD	Density of buildings (c)	2.372	-
GreR	Greening rate of community (c)	0.33	+
Man	Average property management fee per month (yuan/m^2^)	1.153	+
Variables of the Facilities Accessibility			
Hos	Distance to the closest hospital (m)	1144.774	-
Sta	Distance to the closest rail station (m)	1369.524	-
Mar	Distance to the closest market (m)	1081.384	-
Sce	Distance to the closest scenic spot (m)	1075.086	-
Gre	Distance to the closest green space (m)	720.815	-
Wat	Distance to the closest main water source (m)	1150.353	-
Fac	Distance to the closest factory (m)	1578.608	+
Gas	Distance to the closest gas station (m)	1176.205	+
Fun	Distance to the closest funeral facility (m)	3167.635	+
Dum	Distance to the closest dump (m)	9562.531	+
Research Variables			
Residence Type			
Com	Dummy variable, 1 for the residence is pure commercial residence, 0 else	-	+
Res	Dummy variable, 1 for the resettlement residence is included, 0 else	-	-
Ref	Dummy variable, 1 for the housing-reform residence is included, 0 else	-	-
Lim	Dummy variable, 1 for the price-limited residence is included, 0 else	-	-
Aff	Dummy variable, 1 for the affordable residence is included, 0 else	-	-
Ren	Dummy variable, 1 for the public rental residence is included, 0 else	-	-
Pro	Dummy variable, 1 for the private property residence is included, 0 else	-	-
Ten	Dummy variable, 1 for the tenure-only residence is included, 0 else	-	-
Developer Type			
T500	Dummy variable, 1 for the developer is China’s Top 500, 0 else	-	+
MIN	Dummy variable, 1 for the developer origined from Fujian province, 0 else	-	+
MEco	Dummy variable, 1 for the developer established in the market economy period, 0 else	-	+
SPro	Dummy variable, 1 for the developer is sole proprietorship, 0 else	-	-
Lis	Dummy variable, 1 for the developer is listed, 0 else	-	+

**Table 2 ijerph-20-00445-t002:** Regression processes of filtering model variables (N = 1079).

Variables	STE. 1		STE. 2		STE. 3		STE. 4		STE. 5		STE. 6		STE. 7		CFM	
	Coef	t	Coef	t	Coef	t	Coef	t	Coef	t	Coef	t	Coef	t	Coef	t
(Constant)	4.075 ***	3.589	−2.795 ***	−5.746	−2.730 ***	−5.668	−2.581 ***	−5.424	−2.572 ***	−5.399	−2.344 ***	−4.936	−2.332 ***	−4.921	−2.671 ***	−5.621
CityLC1	0.127 ***	7.438	0.102 ***	6.030	0.102 ***	6.177	0.105 ***	6.400	0.098 ***	6.142	0.092 ***	5.757	0.094 ***	5.897	0.096 ***	5.996
CityLC2	0.123 ***	4.247	0.180 ***	6.400	0.177 ***	6.365	0.187 ***	6.837	0.178 ***	6.619	0.179 ***	6.722	0.174 ***	6.538	0.177 ***	6.572
Pop	−0.122 **	−2.972	0.086 ***	3.181	0.090 ***	3.332	0.077 **	2.960	0.082 ***	3.156	0.071 ***	2.749	0.075 ***	2.904	0.086 ***	3.304
PopD	0.408 ***	7.758	0.061 ***	7.764	0.062 ***	7.822	0.060 ***	7.694	0.062***	7.997	0.061 ***	7.930	0.064 ***	8.232	0.063 ***	8.136
EPop	−1.239 ***	−6.666	-	-	-	-	-	-	-	-	-	-	-	-	-	-
GDP	1.367 ***	7.570	0.169 ***	9.026	0.169 ***	9.012	0.174 ***	9.386	0.171 ***	9.239	0.177 ***	9.643	0.174 ***	9.485	0.168 ***	9.096
Age	0.063 ***	3.224	0.069 ***	3.438	0.071 ***	3.541	0.073 ***	3.636	0.075 ***	3.739	0.059 ***	2.938	0.066 ***	3.356	0.081 ***	4.088
PSch	0.270 ***	10.177	0.257 ***	9.507	0.253 ***	9.502	0.253 ***	9.490	0.256 ***	9.610	0.257 ***	9.710	0.248 ***	9.392	0.253 ***	9.477
MSch	0.213 ***	7.650	0.209 ***	7.336	0.213 ***	7.697	0.220 ***	7.987	0.230 ***	8.565	0.229 ***	8.567	0.226 ***	8.480	0.232 ***	8.625
Flo	0.012	0.793	0.007	0.477	-	-	-	-	-	-	-	-	-	-	-	-
BuiD	0.010	0.688	0.012	0.789	-	-	-	-	-	-	-	-	-	-	-	-
GreR	0.039 ***	2.686	0.034 **	2.302	0.032 **	2.193	0.031 **	2.079	0.033 **	2.232	0.022	1.482	0.022	1.513	-	-
Man	0.178 ***	14.356	0.175 ***	13.840	0.178 ***	14.319	0.179 ***	14.437	0.180 ***	14.525	0.161 ***	12.365	0.165 ***	13.061	0.184 ***	15.060
Sta	−0.040 ***	−4.712	−0.053 ***	−6.297	−0.052***	−6.218	−0.055 ***	−6.878	−0.052 ***	−6.636	−0.052 ***	−6.716	−0.052 ***	−6.632	−0.053 ***	−6.717
Hos	−0.027 ***	−2.651	−0.023 **	−2.295	−0.024 **	−2.377	−0.026 **	−2.554	−0.028 ***	−2.795	−0.028 ***	−2.795	−0.028 ***	−2.847	−0.027 **	−2.661
Mar	−0.016 *	−1.824	−0.018 **	−1.997	−0.016 *	−1.901	-	-	-	-	-	-	-	-	-	-
Sce	0.010	0.963	0.005	0.455	-	-	-	-	-	-	-	-	-	-	-	-
Gre	−0.012 *	−1.758	−0.028 ***	−4.117	−0.027 ***	−4.218	−0.027 ***	−4.128	−0.026 ***	−3.999	−0.025 ***	−3.836	−0.026 ***	−4.049	−0.025 ***	−3.894
Wat	−0.044 ***	−5.420	−0.040 ***	−4.883	−0.041 ***	−5.047	−0.040 ***	−4.970	−0.039 ***	−4.853	−0.040 ***	−4.943	−0.040 ***	−4.953	−0.040 ***	−4.920
Fac	0.037 ***	4.047	0.017 **	1.973	0.017 **	1.980	0.016 *	1.810	-	-	-	-	-	-	-	-
Gas	0.018 *	1.812	0.008	0.765	-	-	-	-	-	-	-	-	-	-	-	-
Fun	0.062 ***	4.683	0.043 ***	3.283	0.043 ***	3.317	0.039 ***	3.012	0.039 ***	3.060	0.031 **	2.390	0.033 **	2.540	0.040 ***	3.071
Dum	0.147 ***	8.512	0.138 ***	7.853	0.137 ***	7.858	0.132 ***	7.637	0.135 ***	7.846	0.126 ***	7.346	0.124 ***	7.266	0.135 ***	7.839
Com	-	-	-	-	-	-	-	-	-	-	0.060 ***	4.442	-	-	-	-
Res	-	-	-	-	-	-	-	-	-	-	-	-	−0.065 ***	−4.855	-	-
R^2^	0.682		0.669		0.668		0.667		0.666		0.672		0.673		0.664	
Adj-R^2^	0.675		0.662		0.662		0.662		0.661		0.667		0.668		0.660	

Note: * Significant at the 10% level. ** Significant at the 5% level. *** Significant at the 1% level.

**Table 3 ijerph-20-00445-t003:** Linear regression results of the residence types.

Variables	RESI (N = 1079)			NPCOM (N = 494)		
	Coef	t	Adj-R^2^	Coef	t	Adj-R^2^
Com	0.064 ***	4.751	0.666	-	-	-
Res	−0.068 ***	−5.132	0.667	−0.026	−1.496	0.747
Ref	0.005	0.233	0.659	−0.022	−1.061	0.747
Lim	0.050	0.365	0.659	0.050	0.443	0.746
Aff	−0.010	−0.236	0.659	0.024	0.711	0.746
Ren	−0.017	−0.317	0.659	0.030	0.666	0.746
Pro	−0.020	−0.340	0.659	0.007	0.150	0.746
Ten	0.010	0.245	0.659	0.104 ***	2.985	0.751

Note: *** Significant at the 1% level.

**Table 4 ijerph-20-00445-t004:** Linear regression results of the developer types.

Variables	COM (N = 585)			MCOM (N = 566)			NSCOM (N = 352)		
	Coef	t	Adj-R^2^	Coef	t	Adj-R^2^	Coef	t	Adj-R^2^
T500	0.082 ***	3.779	0.632	0.081 ***	3.704	0.629	0.095 ***	3.730	0.639
MIN	0.051 **	2.047	0.626	0.050 **	1.987	0.622	0.027	0.701	0.625
MEco	0.018	0.363	0.623	-	-	-	-	-	-
SPro	-	-	-	−0.020	−1.014	0.620	-	-	-
Lis	-	-	-	-	-	-	−0.017	−0.525	0.625

Note: ** Significant at the 5% level. *** Significant at the 1% level.

**Table 5 ijerph-20-00445-t005:** Interactive regression results of the residence types (N = 1079).

Variables	COM		RES		REF		LIM		AFF		REN		PRO		TEN	
	Coef	Adj-R^2^	Coef	Adj-R^2^	Coef	Adj-R^2^	Coef	Adj-R^2^	Coef	Adj-R^2^	Coef	Adj-R^2^	Coef	Adj-R^2^	Coef	Adj-R^2^
CityLC1	−0.035	0.667	0.041	0.668	0.024	0.659	−0.254	0.659	−0.057	0.659	−0.320 **	0.661	−0.015	0.659	−0.013	0.659
CityLC2	0.039	0.666	0.000	0.667	−0.115	0.659	−0.254	0.659	−0.126	0.659	−0.082	0.659	−0.028	0.659	−0.066	0.659
Pop	0.027	0.666	−0.038	0.667	0.093	0.659	0.290	0.659	0.032	0.659	0.238	0.660	−0.045	0.659	−0.185	0.660
PopD	−0.003	0.666	0.005	0.667	0.021	0.659	−0.065	0.659	−0.010	0.659	−0.110 **	0.660	0.021	0.659	−0.011	0.659
GDP	−0.076 ***	0.668	0.044	0.668	0.146 ***	0.663	0.738	0.659	0.121	0.659	0.564 ***	0.662	−0.102	0.659	−0.181	0.659
Age	0.107 **	0.668	−0.050	0.668	−0.084 *	0.660	0.050	0.659	0.024	0.659	−0.017	0.659	−0.021	0.659	0.206	0.659
PSch	−0.0183 ***	0.670	0.143 ***	0.669	0.060	0.659	0.050	0.659	−0.010	0.659	−0.447 ***	0.662	−0.020	0.659	0.010	0.659
MSch	−0.087 *	0.667	0.096 **	0.668	0.003	0.659	0.050	0.659	−0.063	0.659	−0.017	0.659	−0.050	0.659	0.010	0.659
Man	0.109 ***	0.672	−0.109 ***	0.672	−0.084 *	0.660	0.424	0.659	−0.056	0.659	−0.056	0.659	0.043	0.659	−0.150 *	0.660
Sta	0.013	0.666	−0.030 *	0.668	0.039	0.659	0.649	0.659	0.005	0.659	0.130 ***	0.661	−0.071	0.659	−0.002	0.659
Hos	0.005	0.666	−0.025 *	0.668	−0.002	0.659	0.267	0.659	0.044	0.659	0.078	0.659	−0.028	0.659	0.086	0.659
Gre	−0.023 **	0.667	0.011	0.667	0.045 *	0.660	0.116	0.659	0.057	0.659	0.056	0.659	0.024	0.659	0.023	0.659
Wat	−0.028 *	0.667	0.025	0.668	0.000	0.659	0.195	0.659	0.043	0.659	0.046	0.659	0.059	0.659	−0.002	0.659
Fun	−0.054 **	0.668	0.041 *	0.668	0.032	0.659	0.050	0.659	0.069	0.659	−0.106	0.659	0.094	0.659	0.033	0.659
Dum	−0.073 ***	0.668	0.072 **	0.669	0.064	0.659	0.427	0.659	0.131	0.659	0.182	0.659	−0.111	0.659	0.002	0.659

Note: * Significant at the 10% level. ** Significant at the 5% level. *** Significant at the 1% level.

**Table 6 ijerph-20-00445-t006:** Interactive regression results of the NPCOM (N = 494).

Variables	RES		REF		LIM		AFF		REN		PRO		TEN	
	Coef	Adj-R^2^	Coef	Adj-R^2^	Coef	Adj-R^2^	Coef	Adj-R^2^	Coef	Adj-R^2^	Coef	Adj-R^2^	Coef	Adj-R^2^
CityLC1	0.067 *	0.749	−0.005	0.746	−0.226	0.746	−0.067	0.746	−0.336 ***	0.752	−0.051	0.746	−0.039	0.750
CityLC2	0.089 *	0.748	−0.175 *	0.748	−0.226	0.746	−0.063	0.746	0.001	0.746	−0.047	0.746	0.052	0.751
Pop	−0.101 *	0.748	0.115 *	0.748	0.258	0.746	0.123	0.746	0.313 **	0.749	−0.063	0.746	−0.158	0.752
PopD	0.002	0.747	0.009	0.746	−0.058	0.746	−0.010	0.746	−0.102 **	0.749	0.029	0.746	0.024	0.751
GDP	−0.021	0.747	0.121 ***	0.752	0.658	0.746	0.030	0.746	0.630 ***	0.755	−0.123	0.747	−0.144	0.751
Age	0.000	0.747	−0.069 *	0.748	0.050	0.746	0.056	0.746	0.030	0.746	−0.035	0.746	0.217	0.751
PSch	0.044	0.747	−0.060	0.747	0.050	0.746	0.024	0.746	−0.520 ***	0.755	0.007	0.746	0.104 ***	0.751
MSch	0.070	0.748	−0.056	0.747	0.050	0.746	−0.149	0.746	0.030	0.746	−0.060	0.746	0.104 ***	0.751
Man	−0.055 *	0.748	−0.006	0.746	0.378	0.746	−0.006	0.746	0.030	0.746	0.085	0.746	−0.030	0.750
Sta	−0.046 **	0.749	0.047 *	0.748	0.579	0.746	0.011	0.746	0.147 ***	0.753	−0.081	0.746	0.008	0.750
Hos	−0.073 ***	0.753	0.003	0.746	0.238	0.746	0.036	0.746	0.093	0.747	−0.003	0.746	0.066	0.751
Gre	−0.039 **	0.750	0.050 **	0.749	0.104	0.746	0.028	0.746	0.030	0.746	0.026	0.746	−0.018	0.751
Wat	0.006	0.747	−0.010	0.746	0.242	0.746	0.053	0.746	0.047	0.746	0.050	0.746	−0.035	0.751
Fun	0.006	0.747	0.012	0.746	−0.446	0.746	0.021	0.746	−0.126	0.747	0.089	0.746	0.055	0.751
Dum	0.061	0.748	0.033	0.746	0.381	0.746	0.057	0.746	0.129	0.747	−0.137 *	0.748	−0.062	0.751

Note: * Significant at the 10% level. ** Significant at the 5% level. *** Significant at the 1% level.

**Table 7 ijerph-20-00445-t007:** Interactive regression results of the developer types (N = 585).

Variables	COM (N = 585)										MCOM (N = 566)						NSCOM (N = 352)					
	T500		MIN		MEco		SPro		Lis		T500		MIN		SPro		T500		MIN		Lis	
	Coef	Adj-R²	Coef	Adj-R²	Coef	Adj-R²	Coef	Adj-R²	Coef	Adj-R²	Coef	Adj-R²	Coef	Adj-R²	Coef	Adj-R²	Coef	Adj-R²	Coef	Adj-R²	Coef	Adj-R²
CityLC1	0.025	0.632	−0.069	0.626	−0.044	0.622	0.016	0.623	0.110	0.624	0.023	0.628	−0.070	0.623	0.018	0.619	−0.009	0.638	−0.124	0.627	0.107	0.626
CityLC2	−0.020	0.632	0.050	0.625	0.222 ***	0.623	−0.054	0.624	−0.013	0.622	−0.021	0.628	0.050	0.622	−0.055	0.620	−0.022	0.638	0.120	0.625	−0.042	0.624
Pop	−0.018	0.632	−0.026	0.625	0.204	0.624	0.019	0.623	−0.068	0.623	−0.029	0.628	−0.022	0.621	0.011	0.619	0.049	0.639	0.060	0.624	−0.066	0.624
PopD	−0.004	0.632	0.025	0.626	−0.020	0.623	−0.011	0.623	0.003	0.622	−0.003	0.628	0.025	0.623	−0.011	0.620	−0.004	0.638	0.048	0.626	−0.004	0.623
GDP	−0.035	0.632	−0.041	0.625	−0.090	0.623	−0.017	0.623	−0.012	0.622	−0.031	0.628	−0.047	0.622	−0.010	0.619	0.023	0.638	−0.092	0.625	−0.037	0.624
Age	0.291 *	0.634	0.053	0.625	0.122	0.623	−0.098	0.624	−0.004	0.623	0.291 *	0.630	0.071	0.622	−0.158	0.621	0.264	0.639	0.027	0.625	−0.017	0.625
PSch	−0.177	0.632	−0.295 **	0.628	−0.048	0.622	0.317 ***	0.632	0.027	0.622	−0.170	0.628	−0.298 **	0.625	0.343 ***	0.629	0.095 ***	0.639	−0.051	0.625	0.246	0.625
MSch	−0.198	0.632	−0.119	0.626	−0.165	0.623	0.022	0.623	0.243	0.623	−0.187	0.628	−0.138	0.623	0.053	0.620	−0.241	0.639	−0.006	0.624	0.260	0.625
Man	0.123 ***	0.637	0.060	0.626	−0.080	0.623	−0.044	0.624	−0.057	0.623	0.130 ***	0.634	0.057	0.622	−0.039	0.620	0.162 ***	0.647	0.015	0.624	−0.070	0.625
Sta	−0.014	0.632	0.056 **	0.628	−0.073	0.623	−0.023	0.624	−0.041	0.623	−0.014	0.628	0.055 *	0.624	−0.020	0.620	−0.019	0.639	0.064	0.626	−0.047	0.625
Hos	−0.038	0.633	−0.015	0.625	−0.052	0.623	−0.007	0.623	−0.043	0.623	−0.035	0.629	−0.018	0.622	−0.006	0.619	−0.038	0.640	−0.025	0.624	−0.031	0.624
Gre	−0.009	0.632	−0.031	0.627	0.031	0.623	−0.012	0.623	0.018	0.623	−0.010	0.628	−0.032	0.623	−0.012	0.620	−0.005	0.638	−0.039	0.625	0.017	0.624
Wat	−0.023	0.632	−0.057 **	0.628	−0.166 *	0.625	−0.025	0.624	0.026	0.623	−0.021	0.629	−0.060 **	0.625	−0.023	0.620	−0.032	0.640	−0.041	0.625	0.028	0.624
Fun	−0.059	0.633	−0.024	0.625	−0.051	0.622	−0.037	0.624	−0.010	0.622	−0.059	0.629	−0.026	0.621	−0.037	0.620	−0.084 *	0.641	−0.059	0.624	−0.027	0.624
Dum	−0.004	0.632	0.009	0.625	−0.194 *	0.624	0.000	0.623	−0.002	0.622	0.002	0.628	0.002	0.621	0.011	0.619	0.038	0.639	0.030	0.624	0.005	0.623

Note: * Significant at the 10% level. ** Significant at the 5% level. *** Significant at the 1% level.

**Table 8 ijerph-20-00445-t008:** Interactive regression results of the developer types (N = 585).

Variables	PCOM (N = 585)												MCOM (N = 566)				NSCOM (N = 352)			
	MCOM				NSCOM				NLCOM				NSCOM				NLCOM			
	MIN		MEco		MIN		SPro		MIN		Lis		MIN		SPro		MIN		Lis	
	Coef	Adj-R^2^	Coef	Adj-R^2^	Coef	Adj-R^2^	Coef	Adj-R^2^	Coef	Adj-R^2^	Coef	Adj-R^2^	Coef	Adj-R^2^	Coef	Adj-R^2^	Coef	Adj-R^2^	Coef	Adj-R^2^
T500	0.087 *	0.638	0.095 ***	0.637	0.089 *	0.638	−0.096 **	0.639	0.085 *	0.639	−0.025	0.637	0.089 *	0.634	−0.096 **	0.635	0.132	0.647	−0.061	0.645
MIN	-	-	0.074 ***	0.637	-	-	0.034	0.636	-	-	−0.107	0.638	-	-	0.033	0.632	-	-	−0.106	0.646

Note: * Significant at the 10% level. ** Significant at the 5% level. *** Significant at the 1% level.

**Table 9 ijerph-20-00445-t009:** Robust test results of linear regression.

Variables	Residence Types						Developer Types									
	RESI (N = 1079)				NPCOM (N = 494)		COM (N = 585)				MCOM (N = 566)				NSCOM (N = 352)	
	Coef	t	Coef	t	Coef	t	Coef	t	Coef	t	Coef	t	Coef	t	Coef	t
(Constant)	−2.100 ***	−5.010	−1.171	−0.405	−1.888 ***	−7.756	−0.256	−0.745	−0.537	−0.986	−0.245	−0.483	−0.558	−0.995	−1.149	−1.525
CityLC1	0.092 ***	5.686	0.111 ***	6.813	0.093 ***	5.077	0.061 **	2.404	-	-	0.040	1.622	-	-	0.076 **	2.440
CityLC2	-	-	0.302 ***	13.313	0.162 ***	5.380	0.181 ***	4.481	0.240 ***	5.978	0.177 ***	4.424	0.219 ***	5.323	0.290 ***	5.918
Pop	0.093 ***	3.956	−0.033	−1.382	-	-	-	-	0.026	0.769	0.020	0.582	0.041	1.186	0.010	0.234
PopD	0.085 ***	12.427	-	-	0.054 ***	6.721	0.045 ***	4.061	0.048 ***	4.074	0.047 ***	4.025	0.052 ***	4.266	0.058 ***	3.890
GDP	0.178 ***	9.880	0.221 ***	11.846	0.215 ***	10.651	0.174 ***	6.286	0.145 ***	5.447	0.187 ***	7.478	0.145 ***	5.207	0.190 ***	5.301
Age	-	-	−0.070 *	1.792	-	-	-	-	-	-	0.115 **	2.135	0.057	1.060	0.200 **	2.008
PSch	0.236 ***	8.768	-	-	0.315 ***	10.868	0.169 ***	3.601	0.160 ***	3.644	0.182 ***	3.922	-	-	-	-
MSch	0.197 ***	7.235	0.287 ***	10.220	0.221 ***	7.325	0.257 ***	5.597	0.253 ***	5.949	0.217 ***	4.777	0.265 ***	5.886	0.231 ***	3.474
Man	0.157 ***	11.998	0.163 ***	12.101	0.106 ***	6.455	-	-	0.217 ***	11.338	0.178 ***	8.548	0.212 ***	10.557	0.209 ***	7.707
Sta	−0.048 ***	−5.983	−0.070 ***	−8.584	−0.047 ***	−4.219	−0.044 ***	−3.665	−0.055 ***	−4.983	−0.048 ***	−4.315	−0.063 ***	−5.511	−0.058 ***	−4.060
Hos	−0.052 ***	−5.446	-	-	−0.014	−1.178	−0.022	−1.360	−0.047 ***	−3.170	−0.050 ***	−3.188	−0.060 ***	−3.931	−0.037 *	−1.913
Gre	−0.039 ***	−6.376	−0.028 ***	−4.021	-	-	−0.037 ***	−3.915	−0.035 ***	−3.930	−0.046 ***	−5.470	−0.035 ***	−3.825	-	-
Wat	−0.029 ***	−3.535	−0.059 ***	−7.016	−0.036 ***	−3.553	−0.035 ***	−2.804	−0.037 ***	−3.265	-	-	−0.040 ***	−3.425	−0.042 ***	−3.048
Fun	-	-	0.016	1.164	0.040 ***	2.850	−0.012	−0.566	-	-	-	-	-	-	-	-
Dum	0.103 ***	5.959	0.088 ***	4.998	0.122 ***	6.181	0.061 **	2.550	0.080 ***	3.499	-	-	0.071 ***	2.997	0.080 ***	2.915
Com	0.080 ***	5.947	-	-	-	-	-	-	-	-	-	-	-	-	-	-
Res	-	-	−0.071 ***	−5.042	-	-	-	-	-	-	-	-	-	-	-	-
T500	-	-	-	-	-	-	0.145 ***	6.535	-	-	0.093 ***	4.226	-	-	0.095 ***	3.782
MIN	-	-	-	-	-	-	-	-	0.056 **	2.246	-	-	0.056 **	2.195	-	-
Ten	-	-	-	-	0.105 ***	3.016	-	-	-	-	-	-	-	-	-	-
Mod	Com		Res		Ten		T500		MIN		T500		MIN		T500	
R²	0.649		0.618		0.747		0.579		0.630		0.624		0.618		0.651	
Adj-R²	0.644		0.613		0.740		0.569		0.621		0.615		0.609		0.637	

Note: * Significant at the 10% level. ** Significant at the 5% level. *** Significant at the 1% level.

**Table 10 ijerph-20-00445-t010:** Robust test results of interactive regression.

Variables	Residence Types																									
	RESI (N = 1079)																									
	Com										Res								Ref		Ren					
	Coef	t	Coef	t	Coef	t	Coef	t	Coef	t	Coef	t	Coef	t	Coef	t	Coef	t	Coef	t	Coef	t	Coef	t	Coef	t
(Constant)	−3.058 ***	−6.386	−0.392	−0.975	−1.426 ***	−6.381	−1.616 ***	−6.612	−2.391 ***	−5.075	−3.529 ***	−7.262	−3.643 ***	−7.201	−3.197 ***	−6.716	−1.222 ***	−6.858	−2.358 ***	−4.776	−491.000	−1.174	−3.541 ***	−7.395	−1.847 ***	−4.419
CityLC1	0.104 ***	6.393	0.101 ***	5.982	0.063 ***	4.104	0.061 ***	3.692	0.090 ***	5.669	0.134 ***	8.210	0.079 ***	4.497	0.108 ***	6.736	0.093 ***	6.315	-	-	0.111 ***	6.569	0.114 ***	6.968	0.095 ***	6.109
CityLC2	-	-	0.286 ***	11.763	0.181 ***	7.173	0.139 ***	5.253	0.177 ***	6.672	-	-	0.082 ***	2.857	-	-	0.220 ***	9.263	-	-	0.315 ***	14.213	-	-	0.224 ***	9.080
Pop	0.113 ***	4.453	−0.026	−1.155	-	-	-	-	0.076 ***	2.940	0.133 ***	5.141	0.180 ***	6.798	0.138 ***	5.423	-	-	0.102 ***	3.776	−0.041 *	−1.723	0.149 ***	5.831	0.044 *	1.857
PopD	0.100 ***	15.156	-	-	0.055 ***	8.059	0.057 ***	7.876	0.062 ***	8.107	0.090 ***	12.716	0.093 ***	11.408	0.090 ***	13.062	0.055 ***	8.088	0.078 ***	10.391	-	-	0.090 ***	12.860	0.058 ***	7.443
GDP	0.173 ***	9.208	0.222 ***	12.219	0.177 ***	10.497	0.237 ***	9.893	0.175 ***	9.600	0.192 ***	10.327	-	-	0.135 ***	7.297	0.198 ***	11.871	0.138 ***	6.453	0.277 ***	15.614	0.124 ***	6.591	0.177 ***	9.988
Age	-	-	0.029	1.376	-	-	0.122 ***	5.715	0.065 ***	3.259	0.070 ***	3.360	0.110 ***	5.226	-	-	0.077 ***	3.864	0.131 ***	5.769	0.065 ***	3.084	-	-	-	-
PSch	0.226 ***	8.303	-	-	0.257 ***	9.672	0.261 ***	9.200	0.331 ***	9.882	0.288 ***	10.489	0.229 ***	7.804	0.198 ***	6.119	0.257 ***	9.696	0.273 ***	9.135	0.316 ***	11.037	0.255 ***	9.286	0.250 ***	9.301
MSch	0.197 ***	7.171	0.282 ***	9.954	0.219 ***	8.129	0.229 ***	7.973	0.226 ***	8.496	-	-	0.277 ***	9.684	0.212 ***	7.780	0.025 ***	6.642	0.243 ***	8.182	-	-	0.213 ***	7.716	0.234 ***	8.662
Man	0.119 ***	6.176	0.160 ***	11.530	0.170 ***	13.078	-	-	0.164 ***	12.748	0.178 ***	11.797	-	-	0.162 ***	12.851	0.165 ***	13.076	-	-	0.194 ***	14.743	0.183 ***	14.986	0.198 ***	16.609
Sta	−0.056 ***	−6.920	−0.069 ***	−8.303	−0.044 ***	−5.580	−0.039 ***	−4.742	−0.053 ***	−6.854	-	-	−0.028 ***	−3.287	-	-	−0.055 ***	−7.126	−0.048 ***	−5.493	−0.073 ***	−8.643	-	-	−0.058 ***	−7.375
Hos	−0.054 ***	−5.650	−0.022 **	−2.024	−0.027 ***	−2.742	−0.018 *	−1.734	−0.030 ***	−2.992	−0.068 ***	−7.000	−0.033 ***	−2.990	−0.062 ***	−6.469	-	-	−0.055 ***	−5.555	−0.028 ***	−2.595	−0.061 ***	−6.272	-	-
Gre	-	-	−0.027 ***	−3.909	−0.032 ***	−5.238	−0.037 ***	−5.767	−0.024 ***	−3.817	−0.047 ***	−7.463	−0.048 ***	−7.106	−0.046 ***	−7.478	−0.020 ***	−3.234	−0.050 ***	−7.519	-	-	−0.047 ***	−7.486	−0.021 ***	−3.277
Wat	−0.046 ***	−5.847	−0.057 ***	−6.739	-	-	-	-	−0.040 ***	−4.979	−0.016 *	−1.933	-	-	−0.022 ***	−2.739	−0.043 ***	−5.371	−0.024 ***	−2.745	−0.060 ***	−7.349	−0.020 **	−2.478	−0.044 ***	−5.389
Fun	0.045 ***	3.486	-	-	0.046 ***	2.848	0.005	0.404	0.030 **	2.336	0.046 ***	3.520	0.059 ***	4.278	0.052 ***	4.005	-	-	0.057 ***	4.072	0.013	0.998	0.061 ***	4.676	-	-
Dum	0.122 ***	6.858	0.124 ***	5.165	0.095 ***	6.013	0.090 ***	5.418	0.123 ***	7.218	0.116 ***	6.365	0.152 ***	7.724	0.114 ***	6.418	0.106 ***	7.000	0.104 ***	5.575	0.097 ***	5.514	0.129 ***	7.212	0.129 ***	7.570
Com	0.079 ***	5.729	0.698 ***	2.683	0.610 ***	3.267	0.739 ***	3.987	0.074 ***	5.424	-	-	-	-	-	-	-	-	-	-	-	-	-	-	-	-
Res	-	-	-	-	-	-	-	-	-	-	−0.088 ***	−6.087	−0.983 ***	−3.468	−0.085 ***	−6.059	−0.078 ***	−5.763	-	-	-	-	-	-	-	-
Ref	-	-	-	-	-	-	-	-	-	-	-	-	-	-	-	-	-	-	−0.882 ***	−2.846	-	-	-	-	-	-
Ren	-	-	-	-	-	-	-	-	-	-	-	-	-	-	-	-	-	-	-	-	−0.984 ***	−2.686	−3.270 ***	−2.799	0.880 **	2.012
Int	0.081 ***	3.174	−0.069 **	−2.420	−0.067 ***	−2.863	−0.098 ***	−3.309	−.183 ***	−3.688	−0.092 ***	−3.317	0.097 ***	3.088	0.128 **	2.356	0.103 **	2.112	0.128 ***	2.687	0.138 ***	2.713	0.529 ***	2.769	−0.104 **	−2.058
Mod	Man		Dum		Fun		GDP		PSch		Man		Dum		PSch		MSch		GDP		Sta		GDP		PopD	
R²	0.643		0.620		0.660		0.615		0.675		0.631		0.591		0.645		0.669		0.578		0.611		0.635		0.655	
Adj-R²	0.638		0.615		0.655		0.610		0.670		0.626		0.585		0.640		0.665		0.573		0.606		0.630		0.651	

Note: * Significant at the 10% level. ** Significant at the 5% level. *** Significant at the 1% level.

**Table 11 ijerph-20-00445-t011:** Robust test results of interactive regression.

Variables	Residence Types												Developer Types					
	NPCOM (N = 494)												PCOM (N = 585)				MCOM (N = 566)	
	Ref		Ren										MIN		T500		MIN	
	Coef	t	Coef	t	Coef	t	Coef	t	Coef	t	Coef	t	Coef	t	Coef	t	Coef	t
(Constant)	−2.109 ***	−9.515	−3.167 ***	−5.061	−3.209 ***	−5.119	−2.263 ***	−9.796	−1.815 ***	−7.432	−3.372 ***	−5.146	−0.463	−1.566	−0.772 ***	−3.013	−0.364	−1.198
CityLC1	0.113 ***	6.302	0.113 ***	6.161	0.119 ***	6.454	0.106 ***	5.789	0.088 ***	4.801	0.132 ***	6.891	0.056 **	2.344	0.080 ***	3.726	0.054 **	2.241
CityLC2	0.176 ***	6.349	0.147 ***	5.343	0.154 ***	5.568	0.140 ***	4.821	0.166 ***	5.471	0.165 ***	5.640	0.249 ***	6.604	0.268 ***	7.556	0.246 ***	6.399
Pop	-	-	0.039	1.176	0.040	1.219	-	-	-	-	0.053	1.528	-	-	-	-	-	-
PopD	0.054 ***	6.691	0.073 ***	7.829	0.072 ***	7.720	0.064 ***	8.303	0.055 ***	6.765	0.070 ***	7.134	0.044 ***	4.257	0.045 ***	4.403	0.044 ***	4.112
GDP	0.197 ***	8.860	0.206 ***	8.911	0.205 ***	8.832	0.215 ***	10.705	0.208 ***	10.321	0.244 ***	10.370	0.158 ***	6.182	0.172 ***	6.743	0.158 ***	5.998
Age	0.082 ***	4.278	0.094 ***	5.048	0.096 ***	5.144	0.095 ***	5.114	-	-	0.075 ***	3.808	-	-	-	-	-	-
PSch	0.334 ***	11.520	0.340 ***	11.732	0.338 ***	11.647	0.343 ***	11.898	0.318 ***	10.819	0.377 ***	12.454	0.419 ***	3.307	0.167 ***	3.869	0.420 ***	3.279
MSch	0.233 ***	7.920	0.227 ***	7.692	0.229 ***	7.750	0.226 ***	7.628	0.221 ***	7.290	-	-	0.223 ***	5.135	0.254 ***	6.003	0.210 ***	4.615
Man	0.102 ***	6.300	0.111 ***	6.995	0.108 ***	6.772	0.106 ***	6.661	0.120 ***	7.459	0.101 ***	6.044	0.223 ***	11.691	0.191 ***	9.722	0.220 ***	11.262
Sta	−0.048 **	−4.520	−0.053 ***	−4.972	−0.048 ***	−4.536	−0.044 ***	−4.105	−0.052 ***	−4.675	−0.072 ***	−6.505	−0.054 ***	−4.880	−0.059 ***	−5.395	−0.056 ***	−4.944
Hos	-	-	-	-	-	-	−0.017	−1.523	−0.014	−1.185	-	-	−0.029 *	−1.919	-	-	−0.032 **	−2.061
Gre	-	-	-	-	-	-	-	-	-	-	-	-	−0.030 ***	−3.486	−0.028 ***	−3.214	−0.030 ***	−3.357
Wat	−0.028 ***	−2.783	-	-	-	-	-	-	−0.038 ***	−3.619	−0.030 ***	−2.719	−0.042 ***	−3.633	−0.043 ***	−3.810	−0.043 ***	−3.696
Fun	0.041 ***	2.926	0.049 ***	3.233	0.046 ***	3.019	0.040 ***	2.898	0.039 ***	2.715	0.041 ***	2.615	-	-	-	-	-	-
Dum	0.135 ***	7.256	0.146 ***	6.206	0.147 ***	6.260	0.121 ***	6.291	0.125 ***	6.327	0.166 ***	6.690	0.086 ***	3.862	0.085 ***	3.890	0.081 ***	3.517
MIN	-	-	-	-	-	-	-	-	-	-	-	-	0.066 ***	2.659	0.080 ***	3.249	0.067 ***	2.636
Ref	−0.811 ***	−3.332	-	-	-	-	-	-	-	-	-	-	-	-	-	-	-	-
Ren	-	-	−1.054 ***	−3.738	0.093 *	1.824	−3.835 ***	−4.202	0.855 **	2.361	0.115 **	2.232	-	-	-	-	-	-
Int	0.122 ***	3.232	0.149 ***	3.813	−0.372 ***	−3.500	0.629 ***	4.218	−0.095 **	−2.255	−0.583 ***	−4.416	−0.301 **	−2.257	0.097 ***	4.461	−0.302 **	−2.242
Mod	GDP		Sta		CityLC1		GDP		PopD2		PSch		PSch		MEco		PSch	
R^2^	0.758		0.755		0.754		0.757		0.745		0.731		0.797		0.643		0.795	
Adj-R^2^	0.751		0.748		0.747		0.750		0.738		0.723		0.636		0.635		0.632	

Note: * Significant at the 10% level. ** Significant at the 5% level. *** Significant at the 1% level.

## Data Availability

Not applicable.

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
