# Peer review of "Types of Resident and Price Distribution in Urban Areas: An Empirical Investigation in China Mainland"

_ijerph, 2022, doi:10.3390/ijerph20010445_

Round 1

Reviewer 1 Report

Considering the prices of housing now, the current literature on the subject should be analyzed. The introduction and literature review should be related to 2019-2022.
The topic of the article does not fit in the profile of the Journal. In my opinion, it would fit in a statistical or econometric journal.
A significant part of the paper is devoted to the analysis and interpretation of linear regression, relationships between data.
The study is extensive and too detailed. The whole can be divided into parts and the results presented in a series of several articles on the subject.
In the current form, the reader feel chaos and confusion.
Why are the variables selected in different years? Population 2020, Per capita 2021?
The commentary after the tables tells what is highly correlated and what is not. There is a lack of explanation and the Authors' own interpretation.

Author Response

  1. Considering the prices of housing now, the current literature on the subject should be analyzed. The introduction and literature review should be related to 2019-2022.

Thank you for your advice. We will add some literature review be related to 2019-2022. 

  1. The topic of the article does not fit in the profile of the Journal. In my opinion, it would fit in a statistical or econometric journal.A significant part of the paper is devoted to the analysis and interpretation of linear regression, relationships between data.

Thank you for your advice. Because the findings of this study are so valuable for urban environment and space, they have been submitted to 'International Journal of Environmental Research and Public Health' in anticipation of publication.

  1. The study is extensive and too detailed. The whole can be divided into parts and the results presented in a series of several articles on the subject.In the current form, the reader feel chaos and confusion.

Thank you for your advice. We will classify the types and degrees to make the reader feel clear.

  1. Why are the variables selected in different years? Population 2020, Per capita 2021?

Thank you for your advice. The study collected panel data on second-hand residential transactions(1079) in Fuzhou City during the March of 2021. Thank you for your reminder. We will amend Population 2020 to Population 2021.

  1. The commentary after the tables tells what is highly correlated and what is not. There is a lack of explanation and the Authors' own interpretation.

Thank you for your advice. We have some data and we will make further explanation on the discussion.

Reviewer 2 Report

Dear Authors,

The article presents an interesting study that examines the influence of factors from an economic perspective on property prices in the city. The authors created a study that revealed different types of residences and developers. The title of the article is very interesting for the general and professional readers. However, before presenting the article, we would recommend a few:

Abstract and Keywords: The abstract is accurate, it contains everything that an abstract should contain. We would recommend shortening it a little if possible. Keywords are appropriately chosen.

Introduction.

- We would recommend inserting at least two sentences between the headings 1. Introduction and 1.1. Background.

- The introduction contains sources, which is fine, but the length of the introduction is too long. We recommend keeping only the essentials and possibly moving some parts to the literature search.

Literature review.

- We recommend inserting at least two sentences between the heading 1.2. literature review and 1.2.1. inluence of residence type.

- We would recommend that the title of the literature review be of the same level and formatting as the Introduction.

- For three or more authors, we would rather recommend writing it as Last name of author et al. (year).

- We would not give level 3 subheadings. Only two.

Methods.

- We would recommend inserting at least two sentences between the 2nd and 2.1 headings.

- Figure 1 should be aligned with the text.

- Table 1 should also be aligned again with the text.

- Again, separate heading 2.4 and 2.4.1 with at least 2 sentences. At the same time, we would recommend giving only 2 levels of heading in this chapter as well.

- We would recommend that the equations should not be underlined

Analysis result: Regarding Analysis result we would recommend the following modifications before publishing the article.

- Again, add at least 2 sentences between headings and subheadings.

- In this section, we would recommend giving only 2 levels of heading.

We would recommend adding at least 2-3 sentences between all headings and subheadings. Once the article has been resolved and edited, we would recommend the article for publication.

Author Response

  1. The article presents an interesting study that examines the influence of factors from an economic perspective on property prices in the city. The authors created a study that revealed different types of residences and developers. The title of the article is very interesting for the general and professional readers. However, before presenting the article, we would recommend a few:Abstract and Keywords: The abstract is accurate, it contains everything that an abstract should contain. We would recommend shortening it a little if possible. Keywords are appropriately chosen.

Thank you for your advice. We will shorten the abstract.

  1.  

- We would recommend inserting at least two sentences between the headings 1. Introduction and 1.1. Background.

- The introduction contains sources, which is fine, but the length of the introduction is too long. We recommend keeping only the essentials and possibly moving some parts to the literature search.

Thank you for your advice. We will insert some sentences between the headings 1. Introduction and 1.1. Background. We will shorten the introduction and move some sentences to the literature review as well.

3.Literature review.

- We recommend inserting at least two sentences between the heading 1.2. literature review and 1.2.1. inluence of residence type.

- We would recommend that the title of the literature review be of the same level and formatting as the Introduction.

- For three or more authors, we would rather recommend writing it as Last name of author et al. (year).

- We would not give level 3 subheadings. Only two.

Thank you for your advice. We will adjust it follow your advice.

4.Methods.

- We would recommend inserting at least two sentences between the 2nd and 2.1 headings.

- Figure 1 should be aligned with the text.

- Table 1 should also be aligned again with the text.

- Again, separate heading 2.4 and 2.4.1 with at least 2 sentences. At the same time, we would recommend giving only 2 levels of heading in this chapter as well.

- We would recommend that the equations should not be underlined

Thank you for your advice. We will adjust it to follow your advice.

  1. Analysis result: Regarding Analysis result we would recommend the following modifications before publishing the article.

- Again, add at least 2 sentences between headings and subheadings.

- In this section, we would recommend giving only 2 levels of heading.

Thank you for your advice. We will adjust it to follow your advice.

  1. We would recommend adding at least 2-3 sentences between all headings and subheadings. Once the article has been resolved and edited, we would recommend the article for publication.

Thank you for your advice. We will adjust it to follow your advice.

Round 2

Reviewer 1 Report

The Authors made the corrections
Text is acceptable for publication